# A GASA Protein Family Gene, *CmGEG*, Inhibits Petal Growth in Chrysanthemum

**DOI:** 10.3390/ijms25063367

**Published:** 2024-03-16

**Authors:** Ziying He, Rui Jiang, Xiaojing Wang, Yaqin Wang

**Affiliations:** 1Guangdong Provincial Key Laboratory of Biotechnology for Plant Development, School of Life Sciences, South China Normal University, Guangzhou 510631, China; hzzyy0022@163.com (Z.H.); 2022205047@stu.njau.edu.cn (R.J.); wangxj@scnu.edu.cn (X.W.); 2Guangdong Laboratory for Lingnan Modern Agricultural, Guangzhou 510642, China; 3Key Laboratory of Landscape Agriculture, Ministry of Agriculture, College of Horticulture, Nanjing Agricultural University, Nanjing 210095, China

**Keywords:** *CmGEG*, GASA protein, petal elongation, GA, chrysanthemum

## Abstract

The diversity in the petal morphology of chrysanthemums makes this species an excellent model for investigating the regulation mechanisms of petal size. However, our understanding of the molecular regulation of petal growth in chrysanthemums remains limited. The GASA (gibberellic acid [GA]-stimulated Arabidopsis) protein plays a significant role in various aspects of plant growth and development. Previous studies have indicated that *GEG* (a gerbera homolog of the gibberellin-stimulated transcript 1 [*GAST1*] from tomato) is involved in regulating ray petal growth by inhibiting cell expansion in gerberas. In this study, we successfully cloned the *GASA* family gene from chrysanthemums, naming it *CmGEG*, which shares 81.4% homology with *GEG*. Our spatiotemporal expression analysis revealed that *CmGEG* is expressed in all tissues, with the highest expression levels observed in the ray florets, particularly during the later stages of development. Through transformation experiments, we demonstrated that *CmGEG* inhibits petal elongation in chrysanthemums. Further observations indicated that *CmGEG* restricts cell elongation in the top, middle, and basal regions of the petals. To investigate the relationship between *CmGEG* and GA in petal growth, we conducted a hormone treatment assay using detached chrysanthemum petals. Our results showed that GA promotes petal elongation while downregulating *CmGEG* expression. In conclusion, the constrained growth of chrysanthemum petals may be attributed to the inhibition of cell elongation by *CmGEG*, a process regulated by GA.

## 1. Introduction

Flowering represents a significant milestone in a plant’s transition from vegetative growth to reproductive growth. The development of the floral organs unfolds in three key stages: floral induction, floral meristem formation, and floral development. Initially, the coordinated interplay of environmental cues (e.g., temperature, photoperiod) and genetic factors triggers the conversion of the apical meristem into the inflorescence meristem [1,2]. Subsequently, auxin accumulates in the inflorescence meristem via polar transport, giving rise to a floral meristem, also known as a floral primordium [3]. Finally, upon the activation of identity genes in the floral meristem, mature floral organs such as sepals, petals, and stamens are generated [4]. Petals play a crucial role in safeguarding the pistil and stamen while also attracting insects to facilitate the successful sexual reproduction of plants [5]. The growth of petals can be divided into two stages. In the early petal development stage, floral primordia undergo cell division and proliferation while organ identity genes are expressed and interact with each other [6]. Subsequently, in the later petal development stage, growth primarily occurs through cell expansion [7].

A growing body of research has demonstrated that the dimensions of petals are governed not solely by genetic factors but also by the levels of phytohormones [8,9,10]. Primarily, auxin is fundamental to petal development [7,11], with genes within the auxin signaling pathway directly influencing petal growth. For example, *Auxin Response Factors 6* (*ARF6*) and *ARF8* are known to progress the elongation of floral organs in *Arabidopsis* [12]. Moreover, in gerbera, gibberellic acid (GA) activates *GhMIF*, and ethylene stimulates *GhEIL1*, both leading to the inhibition of petal elongation [13,14]. In rose, *Rh-PIP1;1* and *Rh-PIP2;1* regulate floral development through the ethylene signaling pathway [15,16]. Beyond auxin, GA, and ethylene, other phytohormones such as brassinosteroids (BRs), abscisic acid, cytokinins, and jasmonic acid are also key regulators of floral growth and development [13,17,18,19,20,21,22]. The application of exogenous BR has been shown to promote petal elongation in both Arabidopsis and gerbera [20,23]. Conversely, petal growth is inhibited by abscisic acid [21]. Cytokinins are crucial for regulating the activity of the inflorescence meristem in Arabidopsis, as evidenced by the formation of larger inflorescence and floral meristems in *ckx3 ckx5* double mutants, a result of increased cell proliferation [24]. The limitation in petal size seen in *opr3* mutants is linked to a deficiency in jasmonate synthesis, which can be partially rescued through the exogenous application of jasmonate [25].

Gibberellic-acid-stimulated *Arabidopsis* (GASA) proteins belong to a class of cysteine-rich peptides (CRPs) characterized by three distinct domains: (1) an N-terminal signal peptide sequence comprising 18–29 amino acid residues, (2) a variable hydrophilic region, and (3) a highly conserved GASA domain including 12 cysteine residues at the C-terminal [26,27]. Importantly, deletion or mutation of the GASA domain has been shown to abolish the function of GASA proteins [28,29]. The research indicates that many members of the *GASA* family are under the regulatory influence of GA, including *GASA1*, *4*, *6*, and *9* in Arabidopsis, *ZmGSL* in maize, *OsGASA* in rice, *Snakin-1* in potato, *GIP1*, *2*, *4*, and *5* in petunias, *CcGASA* in citrus, *PeuGASA* genes in *Populus euphratica*, and *GAST1* in tomato [26,30,31,32,33,34,35,36]. Furthermore, *GASA* family genes have been implicated in intricate hormone signal transduction networks, such as *OsGSR1*, which mediates the crosstalk between the BR and GA signaling pathways in rice [37]. The functions of GASA proteins primarily manifest in various plant growth and development processes, including root, stem, and leaf growth; seed germination; and flower organ development [34,36,38,39,40,41]. For instance, in *Arabidopsis*, *GASA4* regulates floral meristem identity and seed size, *GASA5* acts as a negative regulator of GA-induced flowering and stem growth, *GASA6* influences seed germination by facilitating embryonic axis elongation, and *GASA14* promotes rosette leaf growth [28,29,38,42]. In cereal crops, *OsGASR1* and *OsGASR2* participate in rice inflorescence differentiation, and the downregulation of *OsGSR1* expression impacts leaf and primordial root growth [32,43]. Notably, *Gibberellic-Acid-Stimulated Like-1* (*ZmGSL-1*) is associated with lateral root development in maize, while silencing *Snakin-1* in potato leads to plant dwarfing, a reduced leaf size, and significant alterations in leaf shape [44,45]. Additionally, *GEG* (the *Gerbera hybrida* homolog of the gibberellin [GA]-stimulated transcript 1 [*GAST1*] from tomato) has been identified as an inhibitor during the later stage of ray petal growth, whereas *GhPRGL* promotes ray petal elongation in the early stage [46,47,48]. However, research on the involvement of *GASA*s in the growth and development of Asteraceae remains limited.

*Chrysanthemum morifolium*, renowned as one of the world’s most beloved ornamental flowers, features a capitulum comprising bilaterally symmetrical ray florets and radially symmetrical disc florets, with the ray florets being predominantly female [49,50]. The diverse forms and aesthetic appeal of *C. morifolium* mainly depend on petal shape and size, organ fusion, and floral symmetry. Ectopic expression of the *CmYAB1* and *CmCYC2* genes leads to significant alterations in the petal and inflorescence morphology [51,52,53]. The overexpression of *Cyc2CL-1*, *Cyc2CL-2*, and *CmWUS* influences floral organ development, while transgenic chrysanthemum plants with TCP3-SRDX exhibit modified flower morphology [54,55,56]. Key genes like *CmSVP* and *CmTFL1c* are pivotal to inflorescence formation [57,58]. Moreover, enhanced expression of the brassinosteroid transcription factor *BRI1-EMS-SUPPRESSOR 1* (*CmBES1*) results in increased fusion of the outermost ray florets [59]. Transcriptomic and hormone analyses indicate that the expression of *TCP*s, *bHLH*, *GRXC*, and various hormones likely impact petal growth by altering the cell size and density in ‘Jinba’ [22]. Notably, *CnTCP9* promotes petal cell development and enhances flower size through the GA pathway [60]. Additionally, *CmTCP20*, *CmJAZ1-like*, and *CmBPE2* regulate flower size by influencing cell expansion [61,62].

*Chrysanthemum morifolium* ‘Jinba’ serves as a prevalent subject in the exploration of petal growth and development, with its progression through inflorescence development distinctly classified into the budding stage (BD stage), bud-breaking stage (BB stage), early blooming stage (EB stage), and full blooming stage (FB stage), distinguished by the length of the outermost whorl’s petals [22]. In order to understand the *GASA* gene family’s role, we isolated the homologous gene of *GEG* from ‘Jinba’, named *CmGEG*. We investigated the gene structure, subcellular localization, spatiotemporal expression pattern, and function of *CmGEG*. One of our key findings is that *CmGEG* can regulate petal length by inhibiting cell elongation.

## 2. Results

### 2.1. Cloning and Phylogenetic Analysis of CmGEG

Previously, the *GEG* (AJ005206) gene in gerbera was demonstrated to play a role in shaping the organs and cells by inhibiting cell expansion in the later stages of petal growth [46]. However, it remains unexplored whether its homologous gene, *CmGEG*, functions similarly in chrysanthemum. In this study, we identified a full-length gene of 631 bp with 81.4% homology to *GEG* from the chrysanthemum transcriptome database and designated it as *CmGEG*. The coding sequence (CDS) of *CmGEG* includes a 297 bp open reading frame, encoding a 99-amino-acid protein (Figure 1B).

Analysis on the conserved structural domains of *CmGEG* in NCBI (https://www.ncbi.nlm.nih.gov/cdd/, accessed on 1 June 2020) revealed its membership in the GASA family, characterized by a conserved GASA structural domain (Figure 1A). Alignment analysis of the protein sequence indicated that homologous proteins of CmGEG are widely present in *Arabidopsis* and other species such as *Artemisia annua*, *Tanacetum cinerariifolium*, *G. hybrida*, *Helianthus annuus*, and *Erigeron canadensis*. The similarity between CmGEG and the consensus sequence of other species was found to be 76.7% (Figure 1C), suggesting a high degree of conservation within the GASA family. Phylogenetic analysis of CmGEG and other GASA family members revealed that CmGEG clustered with *Tanacetum cinerariifolium* TcGEG and *Artemisia annua* AaGEG (Figure 1D).

### 2.2. Subcellular Localization of CmGEG

To explore the subcellular localization of CmGEG, we isolated protoplasts from transgenic Arabidopsis plants harboring the engineered vector *pCAMBIA1301-CmGEG-GFP* and the control vector *pCAMBIA1301-GFP*. When expressed independently, the GFP protein was observed in the cell nucleus, membrane, and cytoplasm. However, the CmGEG-GFP fusion protein was found to be localized in the cytoplasm and cell membrane, merged with the red fluorescence emitted by the cell membrane marker mCherry (Figure 2).

### 2.3. The Expression Pattern of CmGEG

The expression pattern of *CmGEG* in various tissues and organs of ‘Jinba’ chrysanthemum was investigated using qRT-PCR to infer its potential functions (Figure 3). The findings revealed that *CmGEG* exhibited high expression levels in the disk and ray florets, while showing low expression in both the young and mature roots (Figure 3A), indicating a more abundant transcriptional abundance in the floral organs. Additionally, the expression of *CmGEG* was examined in four stages of inflorescence development (BD stage, BB stage, EB stage, and FB stage). *CmGEG* expression demonstrated a sharp increase from the BD stage, reaching its peak in the FB stage during inflorescence development (Figure 3B).

### 2.4. CmGEG Is Involved in Chrysanthemum Petal Elongation

To explore the function of *CmGEG*, we initially introduced *CmGEG* into *Arabidopsis* through heterologous transformation. During the seedling stage, the root length in the *CmGEG*-overexpressing (CmGEG-OE) lines was significantly shorter compared to the mock, indicating that *CmGEG* inhibits root elongation in *Arabidopsis*. However, it was observed that the phenotype of the transgenic Arabidopsis plants did not show a significant difference from the mock after seedling establishment (Appendix A).

To further investigate the function of *CmGEG*, we conducted transient transformation assays using chrysanthemum and gerbera petals for functional validation. Initially, a transient overexpression assay was carried out using vacuum infiltration of the *A. tumefaciens* strain containing *CmGEG* under the control of the *CaMV35S* promoter (CmGEG-OE) (Figure 4A). Subsequent qRT-PCR analysis revealed a significantly higher expression level of *CmGEG* in CmGEG-OE compared to the mock, confirming the successful overexpression of *CmGEG* in the petals (Figure 4B). Consequently, we assessed the length and width of the petals and observed that both the length and relative elongation rates in CmGEG-OE were significantly lower than those in the mock (Figure 4A,C,D), while the width remained largely unchanged (Figure 4E). Similarly, transient transformation assays were conducted on detached gerbera ray petals. As depicted in Appendix A, the petals of CmGEG-OE exhibited an increased length and reduced elongation rates compared to the mock, consistent with the findings in chrysanthemum. Additionally, virus-induced gene silencing (VIGS) was employed to suppress *CmGEG* expression (CmGEG-VIGS) (Figure 4F). The expression level of *CmGEG* was significantly decreased in the silenced samples relative to the mock (Figure 4G). As illustrated in Figure 4H–J, the length and relative elongation rate of the petals in CmGEG-VIGS were notably higher than in the mock, with no significant difference observed in the width. Collectively, these results suggest that *CmGEG* functions to inhibit petal elongation in chrysanthemum.

### 2.5. CmGEG Negatively Regulates Petal Size by Limiting Cell Expansion in Chrysanthemum

In order to investigate whether the alteration in petal length was attributed to cell elongation or cell division, we conducted measurements of the size and number of epidermal cells in three distinct regions of the petal (top, middle, and basal) from CmGEG-OE, CmGEG-VIGS, and mock samples after a 6-day cultivation period, following the methodology outlined by Jiang et al. [48] (Figure 5A). As shown in Figure 5B,C, the length of the epidermal cells in the top, middle, and basal regions of the CmGEG-OE petals was significantly shorter than those in the mock. Conversely, the epidermal cells in CmGEG-VIGS exhibited a greater length compared to the mock (Figure 5B,D). Furthermore, no significant differences in the width of the epidermal cells were observed in the transiently transformed petals (Figure 5E,F). Additionally, we observed a significant increase in the number of epidermal cells per unit area in these three regions in CmGEG-OE compared to the mock, while the opposite trend was noted in CmGEG-VIGS (Figure 5G,H). In conclusion, these results suggest that *CmGEG* inhibits petal elongation in chrysanthemum by suppressing cell elongation.

## 3. Discussion

### 3.1. The GASA Protein CmGEG Localize in the Cytoplasm and Cell Membrane

To date, numerous members of the *GASA* family have been identified in various species including Arabidopsis, petunia, potato, rice, cotton, and gerbera, with studies conducted on their structure and subcellular localization [30,34,37,47,63,64]. Accurate subcellular localization can offer valuable insights into the function of proteins. It has been established that GASA proteins localized in the cell wall or extracellular matrix may play a role in cell wall extension or cell division. For instance, GIP2/5 in petunia and GASA5 in *Arabidopsis* were found to be situated in the cell wall. Studies have indicated that *GIP2* is involved in stem elongation, while *GASA5* restricts inflorescence stem elongation by inhibiting cell elongation [34,42].

Some GASA proteins exhibit localization in the cytoplasm and nucleus, resembling the positioning of transcription factors; however, they may not directly regulate downstream genes to carry out their functions. For instance, SlGASA1 in tomato and GmGASA32 in soybean were observed in the cytoplasm and nucleus, where they, respectively, influenced plant height and fruit ripening [64,65]. Apart from cytoplasmic and nuclear localization, OsGSR1 in rice was also detected in the plasma membrane, playing a role in the growth of the primary roots, stems, and leaves [37]. In cotton, GhGASA10-1 was situated in the cell membrane, potentially participating in cell wall development and promoting cell elongation through the secretory pathway [63]. In our investigation, we found that CmGEG is positioned in the cell membrane and cytoplasm (Figure 2), contributing to the regulation of petal elongation by impacting cell expansion (Figure 4 and Figure 5). Therefore, we hypothesize that CmGEG might be involved in the transportation of secretory proteins and cell wall development, ultimately resulting in the inhibition of petal elongation.

Furthermore, the subcellular localization of GASA proteins in certain species has been predicted and analyzed using an online platform [66]. For instance, GASA genes in *Populus trichocarpa* were forecasted to be situated in the cell wall, cell membrane, Golgi apparatus, and nucleus, suggesting that these GASA members may function as membrane proteins involved in transmembrane transport [66]. Despite belonging to the same family, the localization of GASA proteins in different species has been reported to exhibit inconsistencies, potentially attributed to post-translational modifications, electrostatic interactions, covalent bonding with membrane lipids, or attachment/interaction with other proteins [67]. However, further research is required to elucidate the specific underlying reasons.

### 3.2. CmGEG Inhibited Petal Elongation Regulated by GA

Most GASA genes are involved in the regulation of the GA pathway [30,68], with their expression levels being controlled by GA [41,69]. For example, *GAST1* in tomato, *SmGASA4* in *Salvia miltiorrhiza*, *GIP1/2/4/5* in petunia, *OsGSR1* in rice, *GmGASA32* in soybean, *ZmGSL2/4/6* and *ZmGSL9* in maize, *CcGASA7/10/15* in citrus, and *MdGASA18* in apple were upregulated by GA. Conversely, *GASA1/5/9* in *Arabidopsis*, *CcGAS13/16/17* in citrus, *MdGASA8/11/26* in apple, and *Snakin-1* in potato were downregulated [31,33,34,35,36,37,64,70,71].

In gerbera, the expression of *GEG* was upregulated upon the application of GA in the ray floret corollas, and *GEG* was found to play a role in phytohormone-mediated cell expansion [46]. *CmGEG* exhibits a high degree of amino acid sequence homology with *GEG* (Figure 1C), suggesting potential similarities in their biological functions. Subsequent functional investigations revealed that the petal length in the CmGEG-OE plants was shorter compared to the mock controls, whereas the petal length increased in the CmGEG-VIGS plants (Figure 4). Additionally, the CmGEG-OE petal cells were shorter in length and had a higher number of epidermal cells per unit area compared to the mock, while the *CmGEG*-silenced petal cells displayed the opposite characteristics (Figure 5). This study demonstrates that *CmGEG* inhibits petal elongation by influencing cell elongation, similar to *GEG* in gerbera [46]. As mentioned earlier, the expression of most *GASA* family genes is induced by GA. To investigate whether *CmGEG* is also involved in GA regulation, chrysanthemum petals were treated with varying concentrations of GA for different durations. The results indicated that the petal elongation rate peaked after treatment with 5 μM GA, and *CmGEG* expression exhibited a decreasing trend over time, suggesting that *CmGEG* is negatively regulated by GA (Appendix A).

Interestingly, the response of *GASA* genes to exogenous GA_3_ appears to vary depending on the tissue, developmental stage, or treatment duration. For instance, Arabidopsis *AtGASA4* was upregulated by GA in most of the meristem regions but downregulated in the cotyledons and leaves, suggesting that the *AtGASA4* protein is involved in cell division rather than cell elongation [72]. In apple, *MdGASA5* was initially promoted by GA but later inhibited during the flower induction period [70]. Similarly, *GsGASA1* expression was inhibited in the roots but stimulated in the leaves upon GA treatment [40]. Moreover, it is worth noting that not all *GASA* family genes are regulated by GA. For example, *AtGASA2/3/10/12/14/15* in *Arabidopsis*, *FaGAST2* in strawberry, and *GhGASA10-1* in cotton showed no response to GA treatment [30,39,63].

## 4. Materials and Methods

### 4.1. Plant Materials and Growth Conditions

In this study, a cultivar of chrysanthemum known as ‘Jinba’ was utilized. The seedlings were cultivated under greenhouse conditions, which included natural light, a day/night temperature of 26/20 °C, 70% relative humidity, and a substrate composed of a 3:1 (*v*/*v*) mixture of vermiculite and Soilrite.

The seeds of Col-0, an ecotype of *Arabidopsis thaliana*, were soaked in 75% ethanol and shaken for 2 min, sterilized with 2% sodium hypochlorite for 6 min, and finally washed with sterile water 6 times. The seeds of *Arabidopsis thalian* were plated on Murashige and Skoog (MS) medium. Following germination, the plates were moved to a tissue culture room maintained at 22 ± 2 °C with a 16 h light/8 h dark cycle and a relative humidity ranging from 60% to 80%. After 7 days of cultivation, the seedlings were transplanted into a soil medium under controlled conditions at 24 ± 2 °C.

### 4.2. Cloning and Phylogenetic Analysis of CmGEG

The open reading frame (ORF) sequence of *CmGEG* was isolated from the cDNA of ‘Jinba’ using the gene-specific primers CmGEG-F1/R1 (Appendix A). The ORF of the *CmGEG* sequence was then translated into amino acids using DNAMAN software 9.0. Prediction of the conserved structural domains from the protein sequence was undertaken using NCBI databases. Homology analysis of CmGEG was conducted using BlastP, and a phylogenetic tree was constructed based on protein sequences utilizing MEGA 6.0 software and using the neighbor-joining (NJ) method [73].

### 4.3. Ectopic Expression of CmGEG in Arabidopsis

The full-length ORF of *CmGEG*, excluding the stop codon, was amplified using the primer pair CmGEG-F2/R2 (Appendix A). Subsequently, the ORF was inserted into the *BamH-I* and *Sal-I* sites of the pCAMBIA1301-GFP vector to create the recombinant expression vector pCAMBIA1301-CmGEG-GFP. Following this, the pCAMBIA1301-CmGEG-GFP fusion plasmid and the pCAMBIA1301-GFP empty vector were introduced into *Agrobacterium tumefaciens* strain GV3101 using the freeze–thaw technique. These were then transformed into the Col-0 ecotype utilizing the floral dip transformation method, as previously described [74]. The seeds of the transgenic plants were screened on 1/2 MS medium containing 50 μg/mL hygromycin, and the T3 transgenic plants were identified using RT-PCR with the primer pair CmGEG-F1/R1 (Appendix A).

### 4.4. Subcellular Localization of CmGEG

Protoplasts containing the pCAMBIA1301-CmGEG-GFP fusion plasmid and the pCAMBIA1301-GFP empty vector were extracted from the leaves of 4-week-old Arabidopsis plants, following the previously established protocol [48,75]. To label the cell membrane, the cell membrane localization marker mCherry (a plasmid with the cell membrane localization signal fused to the mCherry protein) was co-transfected into the Arabidopsis protoplasts. Fluorescence analysis was conducted using a laser confocal scanning microscope (LSM800, Carl Zeiss, Oberkochen, Germany) approximately 12 h post-transformation.

### 4.5. Transient Transformation of the Petals

Transient overexpression and virus-induced gene silencing (VIGS) were conducted in the petals following the previously established protocols [13,14]. The open reading frame (ORF) of *CmGEG* was utilized to construct vectors for overexpressing *CmGEG* (pCanG-CmGEG) and silencing *CmGEG* (pTRV2-CmGEG). Subsequently, the *A. tumefaciens* strain C58C1 was transformed with pCanG-CmGEG, pCanG, pTRV1, pTRV2-CmGEG, and pTRV2, respectively. These *A. tumefaciens* strains were cultured in 5 mL Luria–Bertani (LB) medium supplemented with 50 mg/mL kanamycin and 50 mg/mL rifampicin at 220 rpm and 28 °C overnight. The cultures were then diluted 1:100 (*v*/*v*) into 100 mL LB medium containing 20 μM acetosyringone (AS) and 10 mM 2-(N-morpholinyl) ethanesulfonic acid (MES) and grown overnight. Upon reaching an absorbance (OD_600_) of 1.0–1.5, the bacterial cultures were harvested via centrifugation at 4000× *g* for 10 min and resuspended in an infiltration buffer (200 µM AS, 10 mM MES, and 10 mM MgCl_2_, pH = 5.6) to a final OD_600_ of approximately 1.2. Subsequently, the *A. tumefaciens* cultures carrying the vectors pCanG-GEG, pCanG, pTRV2-CmGEG/pTRV1, and pTRV2/pTRV1 were incubated separately in the dark at room temperature for 4–6 h.

The petals were isolated from the inflorescence, submerged in sterile water, wrapped in gauze, and then subjected to different *A. tumefaciens* infiltration solutions under a vacuum of −0.09 MPa for 10 min, followed by a gradual return to normal atmospheric pressure within 2 min, following the procedures outlined in previous studies [14,48]. Subsequently, the infiltrated petals were rinsed with sterile distilled water (ddH_2_O) and placed in culture dishes lined with filter papers. After a 3-day incubation at 4 °C in continuous darkness, the petals were moved to a growth room set at 23~25 °C under a 16 h light/8 h dark cycle with a relative humidity of 50–60%. Throughout the cultivation period, petals were randomly sampled on alternate days for the gene expression analyses.

### 4.6. Measurement of the Petal Size and Cell Size

Daily photographs of the petals were captured using a digital camera (Nikon, D7200, Tokyo, Japan). The petal length, cell lengths, and width were measured following the previously established protocols [25]. Approximately 1 mm^2^ sections from the top, middle, and basal regions of the petals were excised and stained with 0.1 mg/mL propidium iodide (PI) at room temperature for 30 min, as outlined in previous studies [62]. The morphology of the upper epidermal cells was examined and documented using a laser confocal microscope (LSM800, Carl Zeiss, Oberkochen, Germany). Each treatment involved three biological replicates of the petals, with approximately 9 petals observed for each biological replicate. We measured 36 cells per sample to observe the cell size of the petal. The petal length/width, cell length/width, and cell density per unit area were quantified using ImageJ 1.38e software.

### 4.7. RNA Extraction and Quantitative RT-PCR (qRT-PCR)

Total RNA was extracted from various tissues (such as the roots, stems, leaves, sepals, disk florets, and ray florets) and from inflorescences in four developmental stages (BD, BB, EB, and FB) using the Eastep^®^ Super Total RNA Extraction Kit (Promega, Guangzhou, China, Code No. LS1040). Subsequently, the RNA was reverse-transcribed into cDNA using the ReverTra Ace qPCR RT Kit (TOYOBO, Code No. FSQ-301) following the manufacturer’s instructions. Quantitative real-time PCR (qRT-PCR) was carried out using the CFX96 Touch^TM^ Real-Time PCR Detection System (Bio-Rad Laboratories, Shanghai, China) and using 2× RealStar Green Fast Mixture (GenStar, Guangzhou, China, Code No. A301-01) according to the manufacturer’s protocol. The primer pairs (qCmGEG-F8/qCmGEG-R8) listed in Appendix A were utilized for the qRT-PCR analyses, with *Elongation Factor 1α* in chrysanthemum (*CmEF1α*) (GenBank: AB548817.1) serving as the internal reference gene, as described previously [76]. Each sample underwent three biological replicates and three technical replicates, and the data were analyzed using the 2^−∆∆Ct^ method [77].

### 4.8. Statistical Analysis

All the statistical data in this study were subjected to three biological replications, and statistics are presented as mean ± standard deviation. The data significance analysis was performed using Duncan’s algorithm in SPSS 17.0 according to one-way analysis of variance.

## 5. Conclusions

Our study revealed that *CmGEG* is localized in the cytoplasm and cell membrane, with its expression levels significantly increasing in correlation with floral growth and development, peaking during the full blooming (FB) stage. Ectopic transformation experiments demonstrated that *CmGEG* inhibits root elongation in Arabidopsis. Subsequent transient transformation assays in chrysanthemum indicated *CmGEG* was involved in the regulation of petal elongation. Measurements of the petal epidermal cell length, width, and count revealed that *CmGEG* primarily inhibits petal elongation by reducing cell length. Overall, our research suggests a potential pathway for understanding the molecular mechanisms behind petal elongation in chrysanthemum. However, the possibility of *CmGEG* acting in concert with other genes to regulate petal elongation requires further investigation.

## Figures and Tables

**Figure 1 ijms-25-03367-f001:**
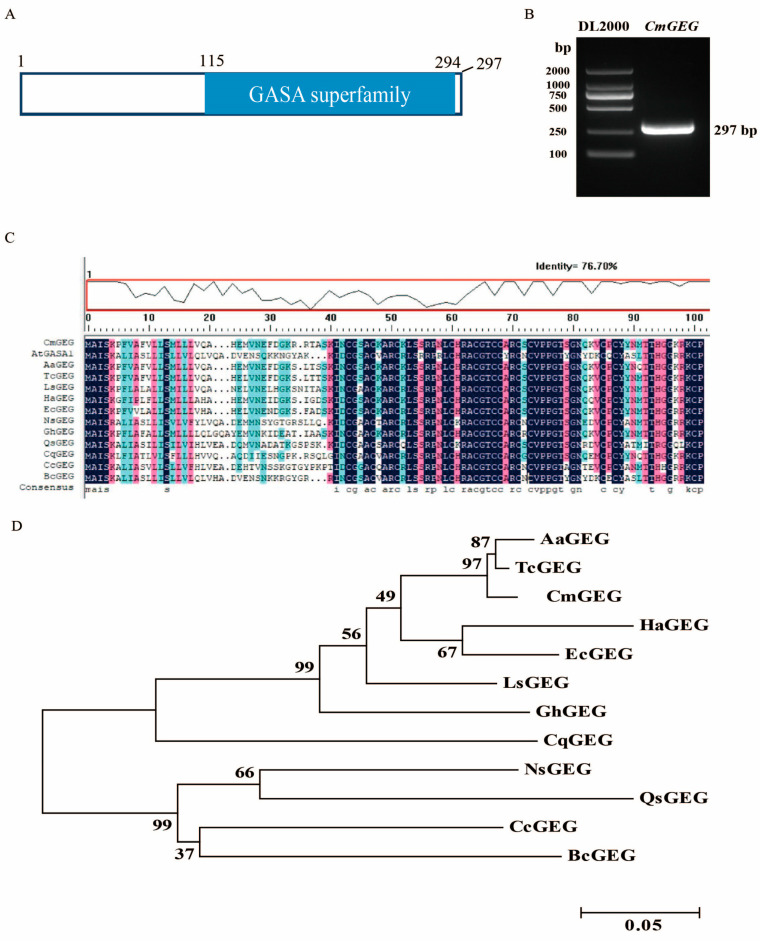
Gene structure and phylogenetic analyses of *CmGEG*. (**A**) Analysis of the GASA domain of *CmGEG*. Conserved structural domain of *CmGEG* was predicted by using NCBI conserved domain database (https://www.ncbi.nlm.nih.gov/cdd/, accessed on 1 June 2020). The GASA superfamily of *CmGEG* was located between nucleotides 115 and 294 of the gene. (**B**) Cloning of *CmGEG*. (**C**) Amino acid alignment of CmGEG proteins in various plant species. The similarity between CmGEG and consensus sequence of other species was 76.7%. (**D**) Phylogeny of the *GASA* family genes in different species. The bootstrap values shown indicate the robustness of each branch. The scale bar corresponds to 0.05 substitutions per site. Amino acid sequences were used for the amino acid alignment and phylogeny analysis. At, *Arabidopsis thaliana*; Aa, *Artemisia annua*; Tc, *Tanacetum cinerariifolium*; Ls, *Lactuca sativa*; Ha, *Helianthus annuus*; Ec, *Erigeron canadensis*; Ns, *Nyssa sinensis*; Gh, *Gerbera hybrida*; Qs, *Quercus suber*; Cq, *Chenopodium quinoa*; Cc, *Citrus clementina*; Bc, *Brassica carinata*.

**Figure 2 ijms-25-03367-f002:**
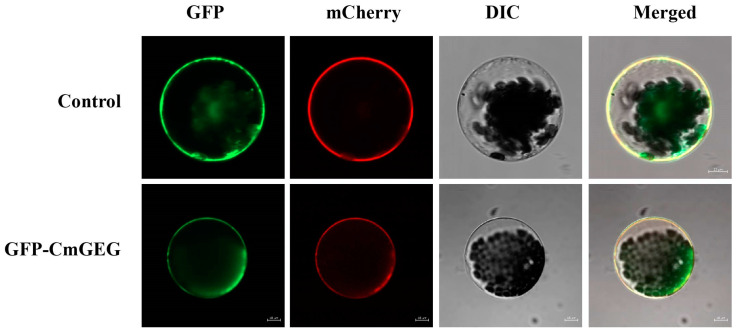
Subcellular location of CmGEG. Subcellular localization of the CmGEG protein in Arabidopsis protoplasts. GFP protein driven by the *35S* promoter was transformed as a control. GFP, images taken in the green fluorescence channel; mCherry, cotransformed as cell membrane marker. DIC, images taken in bright light; Merged, both overlay plots. Scale bar = 10 µm.

**Figure 3 ijms-25-03367-f003:**
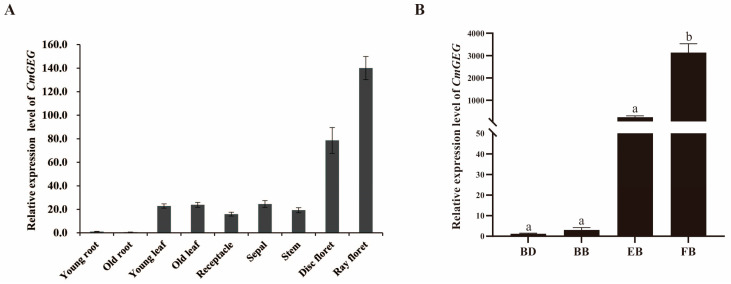
Expression patterns of *CmGEG* in chrysanthemum. (**A**) Expression patterns of *CmGEG* in different tissues and organs. (**B**) Expression pattern of *CmGEG* in the four floral development stages. The four stages of ‘Jinba’ floral development: budding stage (BD stage), bud-breaking stage (BB stage), early blooming stage (EB stage), full blooming stage (FB stage). Values are the mean ± SD from three biological replicates. Tukey’s (honestly significant difference) HSD test was used to determine significant differences; letters above bars indicate significant differences (*p* < 0.05).

**Figure 4 ijms-25-03367-f004:**
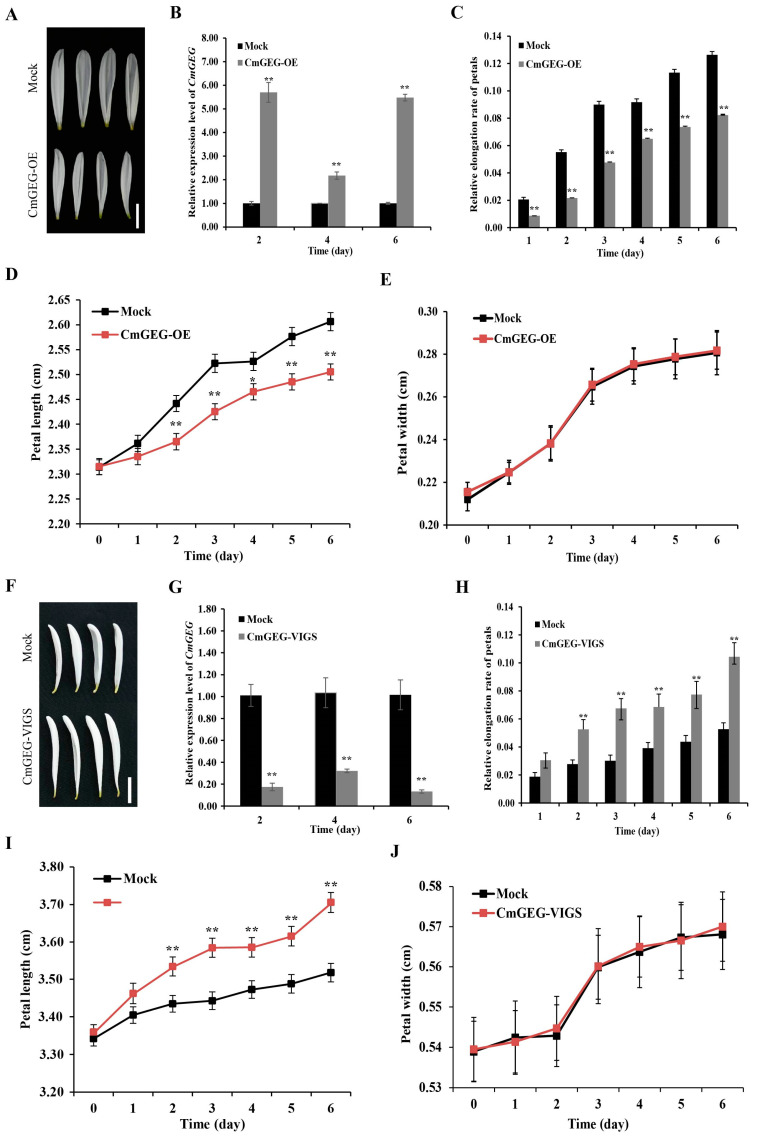
*CmGEG* inhibits petal elongation of chrysanthemum. Petal phenotypes of *CmGEG*-OE (**A**) and *CmGEG*-VIGS (**F**) after 6 days of culturing. Expression levels of *CmGEG* in the mock and *CmGEG*-OE (**B**) and the mock and *CmGEG*-VIGS are shown in (**G**), respectively. (**C**,**H**) Relative elongation rate of petals in each treatment. Time-course dynamics of petal length in *CmGEG*-OE (**D**) and *CmGEG*-VIGS (**I**) after transformation. Petal width of *CmGEG*-OE (**E**) and *CmGEG*-VIGS (**J**). Scale bars are 1 cm in (**A**,**F**). Each observation was performed using at least three biological replicates Tukey’s HSD: ** p* < 0.05, *** p* < 0.01.

**Figure 5 ijms-25-03367-f005:**
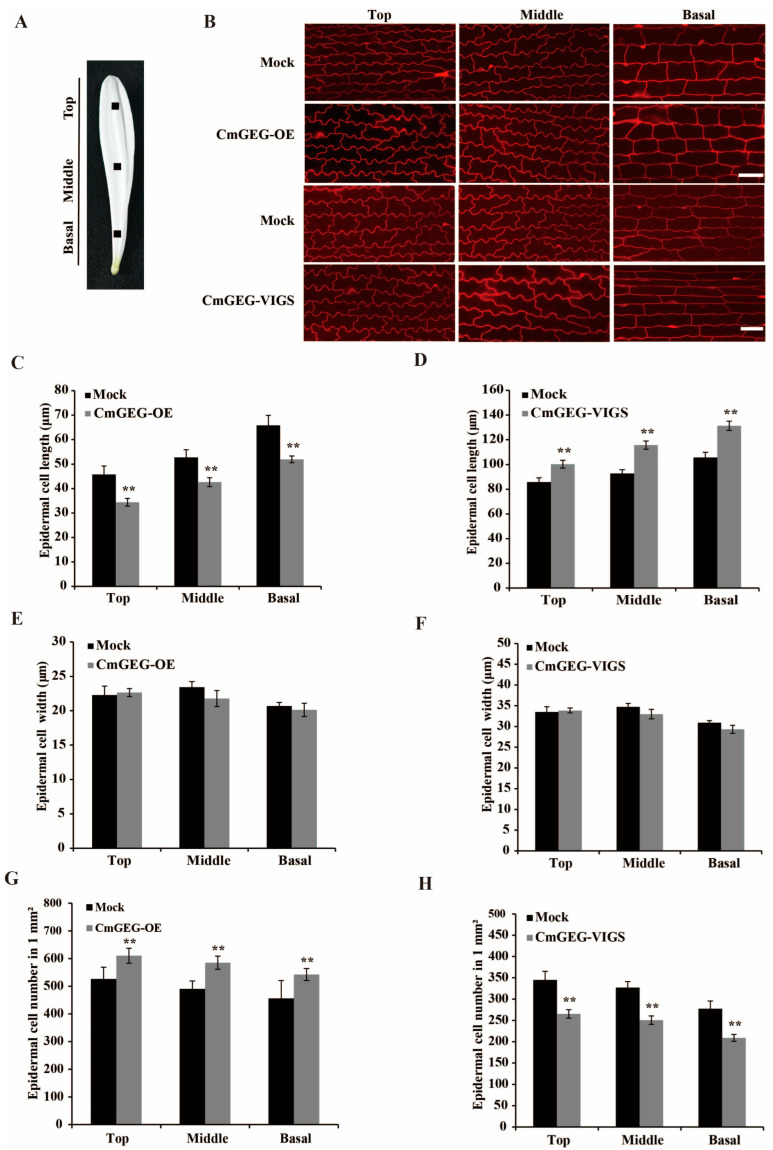
*CmGEG* inhibits the expansion of petal cells. (**A**) Blocks (each 1 mm^2^) at the center of the basal, middle, and top regions of ray petals were sampled for the morphological characterization of petal cells. (**B**) Morphological characterization of adaxial epidermal cells in the basal region of control, CmGEG-OE, and CmGEG-VIGS petals. Epidermal cell length of CmGEG-OE (**C**) and CmGEG-VIGS (**D**) petals in the top, middle, and basal regions. Epidermal cell width of CmGEG-OE (**E**) and CmGEG-VIGS (**F**) petals in the top, middle, and basal regions. Cell number of epidermal cells per unit area (1 mm^2^) of CmGEG-OE (**G**) and CmGEG-VIGS (**H**) petals in the top, middle, and basal regions. Values are the mean ± SD of three biological replicates. Tukey’s HSD: ** *p* < 0.01. Scale bars are 1 μm in (**B**).

## Data Availability

The data are contained within the article and Appendix A.

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
