# Peer review of "A GASA Protein Family Gene, CmGEG, Inhibits Petal Growth in Chrysanthemum"

_ijms, 2024, doi:10.3390/ijms25063367_

Round 1

Reviewer 1 Report

Comments and Suggestions for Authors

The reviewed paper is devoted to the identification of a gene controlling petal length in Chrysanthemum and its functional analysis. The authors applied modern methods and the results obtained are reliable and of interest for further dissection of this complex trait. I am especially pleased to highlight that the paper is quite concise and generally easy to read and understand. I have only minor concerns about this paper.
1. Both language and style require some elaboration, preferentially by a native speaker. I have left some questions and comments in the file (see attached) which may be helpful to revise this paper but even more thorough language and style check is needed. For example, it is important to check where scientific (Latin) and gene names should be italicized and where not.
2. I would suggest that the authors should reduce their Discussion section in favour of really discussing their own results in light of what is known from previously published works. At the moment, some paragraphs of the Discussion can be either omitted or moved to the Introduction as they do not contain any novel findings but revise earlier works in this field. This may help make this paper even shorter.
After taking into account these and other comments (see attached file), this paper can be recommended for publication. I wish the authors good luck.

Comments on the Quality of English Language

This paper is better to be reviewed by a native speaker or a specialized language service to improve some points.

Author Response

The reviewed paper is devoted to the identification of a gene controlling petal length in Chrysanthemum and its functional analysis. The authors applied modern methods and the results obtained are reliable and of interest for further dissection of this complex trait. I am especially pleased to highlight that the paper is quite concise and generally easy to read and understand. I have only minor concerns about this paper.

Response:

Thanks for your accurate summary of my manuscript. Your helpful comments were taken into consideration.

Major points:

1. Both language and style require some elaboration, preferentially by a native speaker. I have left some questions and comments in the file (see attached) which may be helpful to revise this paper but even more thorough language and style check is needed. For example, it is important to check where scientific (Latin) and gene names should be italicized and where not.

Response:

Thanks for your comments. We have elaborated the language and style of this article, and since the modified parts are too many and messy, we have directly modified your comments on this basis. We read the relevant literature and found that the “Arabidopsis” does not require italics, whereas the “Arabidopsis thaliana” does. The scientific (Latin) and gene names should be italicized have been modified according to the submission details of this journal and were marked in red, as shown in lines 73, 90, 103, 367. We have also revised other opinions in the article, please check them.

2. I would suggest that the authors should reduce their Discussion section in favour of really discussing their own results in light of what is known from previously published works. At the moment, some paragraphs of the Discussion can be either omitted or moved to the Introduction as they do not contain any novel findings but revise earlier works in this field. This may help make this paper even shorter.

Response:

Your question is very valuable. We have revised the discussion section, please see lines 230-307.

3. I think reproductive growth in angiosperms includes flowering as a key event. Otherwise stated, flowering is not a mere indicator.

Response:

Thank you very much for your suggestions. We have modified the expression of this sentence, please see lines 34-35.

4. At the first mention, please give genes' names in full. In plant genetics, small italics means mutations. In the context of this sentence, functional alleles, not mutations, are in focus. They should be therefore designated as ARF6 and ARF8.

Response:

Thanks for your comments. We are sorry that we missed the description of genes' names in full when first mentioned. In this sentence, we focused on the function of the gene and have modified the description of arf6 and arf8, as shown in lines 51-52.

5. Is it really twelve Cys residues one by one? If not, then probably 'consisting' should be replaced by 'including' or 'enriched by'.

Response:

Thanks for your suggestions. We are sorry for using the wrong description in “a highly conserved GASA domain consisting of 12 cysteine residues at the C-terminal”. The GASA domain of twelve Cys residues is not one by one and we have removed the “consisting’’, which can be seen in lines 68-69.

6. Please rephrase this sentence to avoid beginning it with a figure. As different species have different sequences of orthologous genes, I cannot understand which comparison is characterized with 76.7% of similarity. Is it the similarity between CmGEG and the consensus sequence of other species? Please specify this.

Response:

Thank you for pointing out the deficiencies in our manuscript. We are very sorry that we made a mistake here. As you said, the similarity between CmGEG and the consensus sequence of other species was 76.7%. We have corrected the description, which can be seen in lines 131-132, 141-142.

7. Please specify what the scale bar length is in Figure 4A and F. In Figure 4I, please add extra digit after in all lengths (3.40, 3.50 etc. instead of 3.4 and 3.5).

Response:

Thanks for your comments. In our article, we specified scale bars in Figure 4A and F, the original was “Scale bars are 1 cm in (A) and (F)”. Please see line 214 and we marked in red. We are very sorry that we made a mistake in Figure 4I. Now, we have added extra digit after in all lengths in Figure 4.

8. How to sterilized seeds of Arabidopsis thaliana (line 316)?

Response:

Thanks for your comments. The method of sterilization for Arabidopsis seeds are as follows: Seeds of Arabidopsis thaliana were soaked in 75% ethanol and shaken for 2 minutes, then sterilized with 2% sodium hypochlorite for 6 minutes, and finally washed with sterile water for 6 times. We have added in lines 316-318.

9. What units is it expressed in 1.0-1.5 in the sentence of “When the absorbance (OD600) of the cultured bacteria reached 1.0-1.5” (line 359)?

Response:

Thanks for your comments. "OD600" is used to refer to a spectrophotometer method that is used to help estimate the concentration of bacteria or other cells in a liquid sample. The OD600 is usually ununitless in the process used.

10. How many cells per sample did you measure?

Response:

Thanks for your comments. We measured 36 cells per sample to observe the cell size of petal and three biological replicates of petals were taken for each treatment. We have added in lines 394-395.

11. Please double check if all scientific (generic and specific Latin) and gene names are italicized in References.

Response:

Thanks for your comments. We check all scientific (generic and specific Latin) and gene names in References. The places that need to be modified are marked in red, please see lines 467-468.

Reviewer 2 Report

Comments and Suggestions for Authors

The manuscript with the title “A GASA Protein Family gene, CmGEG, Inhibits Petal Growth in Chrysanthemum” elucidated CmGEG in chrysanthemum flower development. Main findings were that CmGEG had more abundant transcriptional levels in floral organs, particularly in later stages of petal growth (regulation of petal elongation by affecting cell expansion), and was located in cell membrane and cytoplasm. The manuscript has elements of novelty.

Abstract provides a good summary of the work.

Introduction gives a good background for the research.

Lines 106-117 paragraph should express the aim and objectives (objectives as steps proposed to reach the aim). Such a structure will be very helpful for readers to have a clear structure and preparing them for what they are about to read. It should not contain conclusions (Line 116-117).

Results

Figure 5B – caption should explain the size of the scale from the image in microns.

Discussion is relevant for the research conducted.

Conclusions – authors decided do not have a conclusions section. I suggest to insert a conclusions section and to structure this in a way to mirror the structure of aim and objectives given at the ed of the introduction. Each objective shall have its conclusion, hence all objectives covering the entire results. This way readers can easily find what they are looking for in the paper.

References are on topic.

Best regards.

Comments on the Quality of English Language

Fine English style and grammar improvements are recommended.

Author Response

The manuscript with the title “A GASA Protein Family gene, CmGEG, Inhibits Petal Growth in Chrysanthemum” elucidated CmGEG in chrysanthemum flower development. Main findings were that CmGEG had more abundant transcriptional levels in floral organs, particularly in later stages of petal growth (regulation of petal elongation by affecting cell expansion), and was located in cell membrane and cytoplasm. The manuscript has elements of novelty.

Response:

Special thanks to you for your good summary of our manuscript.

Major points:

1.Introduction gives a good background for the research.

Lines 106-117 paragraph should express the aim and objectives (objectives as steps proposed to reach the aim). Such a structure will be very helpful for readers to have a clear structure and preparing them for what they are about to read. It should not contain conclusions (Line 116-117).

Response:

Thank you for pointing out the deficiencies in our manuscript. We have elaborated the language and style of this article. We have modified the contents in the fifth paragraph of the introduction (lines 108-116).

2.Results:Figure 5B – caption should explain the size of the scale from the image in microns.

Response:

Thank you for pointing out the mistake in our manuscript. We are very sorry that we made a mistake here. Now, we have changed it to the correct one in line 266 and we marked in blue.

3.Discussion is relevant for the research conducted.

 Conclusions – authors decided do not have a conclusions section. I suggest to insert a conclusions section and to structure this in a way to mirror the structure of aim and objectives given at the ed of the introduction. Each objective shall have its conclusion, hence all objectives covering the entire results. This way readers can easily find what they are looking for in the paper.

Response:

Your question is very valuable. We are sorry that we missed the section of conclusion. We have inserted a conclusions section, as shown in line 404-415.